# Properties of repression condensates in living *Ciona* embryos

Nicholas Treen[1], Shunsuke F. Shimobayashi[2], Jorine Eeftens [2], Clifford P. Brangwynne [1,2,3] & Michael Levine [1,4✉]

Recent studies suggest that transcriptional activators and components of the pre-initiation complex (PIC) form higher order associations—clusters or condensates—at active loci. Considerably less is known about the distribution of repressor proteins responsible for gene silencing. Here, we develop an expression assay in living *Ciona* embryos that captures the liquid behavior of individual nucleoli undergoing dynamic fusion events. The assay is used to visualize puncta of Hes repressors, along with the Groucho/TLE corepressor. We observe that Hes.a/Gro puncta have the properties of viscous liquid droplets that undergo limited fusion events due to association with DNA. Hes.a mutants that are unable to bind DNA display hallmarks of liquid–liquid phase separation, including dynamic fusions of individual condensates to produce large droplets. We propose that the DNA template serves as a scaffold for the formation of Hes condensates, but limits the spread of transcriptional repressors to unwanted regions of the genome.

[1] Lewis-Sigler Institute for Integrative Genomics, Princeton University, Princeton, NJ, USA. [2] Department of Chemical and Biological Engineering, Princeton University, Princeton, NJ, USA. [3] Howard Hughes Medical Institute, Chevy Chase, MD, USA. [4] Department of Molecular Biology, Princeton University, Princeton, NJ, USA. ✉email: msl2@princeton.edu

There is emerging evidence that liquid–liquid phase separation processes (LLPS) represent a major mechanism of cellular organization[1]. Many examples of intracellular condensates have been documented, including speckles and other nuclear bodies contained within the nucleus[2]. The best characterized of these is the nucleolus[3], the largest nuclear compartment and site of ribosomal RNA transcription by RNA Polymerase I (Pol I)[4]. Because rRNAs are the predominant transcriptional output of the nucleus, it has been suggested that the nucleolus represents an extreme example of the same overall regulatory principles underlying Pol II-mediated transcription[2]. Indeed, several recent studies provide evidence for clustering of transcription complexes at active loci, particularly components of the pre-initiation complex (PIC) such as Mediator and Pol II[5–7]. A key conclusion of these studies is that the DNA template plays an important role in the formation of PIC condensates, particularly sequence elements contained within distal enhancers and their target promoters[8].

Since most previous studies of LLPS have focused on activators and co-activators such as components of the Mediator complex, we decided to examine the properties of transcriptional repressors. Towards this goal we developed a visualization assay in living *Ciona* embryos since they possess a number of favorable features to explore the cell biology of LLPS, including large, well-defined cells and nuclei, rapid mitotic cycles and small cell numbers that permit the use of sophisticated live imaging methods[9]. Particular efforts focused on the Hes/Hairy family of sequence-specific transcriptional repressors since they have been implicated in a variety of important developmental processes such as segmentation of the *Drosophila* embryo and somitogenesis in vertebrates[10]. These proteins recognize specific DNA sequence motifs via a basic helix-loop-helix domain, but instead of recruiting coactivators, they interact with the Groucho/TLE (Gro) family of corepressor proteins through a short C-terminal

peptide motif, WRPW[11,12]. Gro contains a series of WD40 repeats that are known to mediate the formation of Hes-Gro oligomers, which establish stable and dominant silencing of gene activity[13]. WD40-containing corepressors are highly conserved in evolution, from yeast to humans[14].

Here we present evidence that Hes proteins form spherical puncta that display major hallmarks of LLPS, including the ability to fuse, recover from photobleaching, and dissolve and reassemble during mitosis. These properties depend on Hes-Gro interactions and an intact matrix of F-actin. The analysis of Hes mutants suggests that DNA limits fusions of Hes condensates. We propose that this inhibition is essential to restrict transcriptional repressors to local sites of gene silencing within the genome.

## Results

***Ciona* condensate assay.** We developed an expression assay in *Ciona* embryos to visualize phase separated condensates. It is ideal for such studies due to the ease of expressing fluorescent fusion proteins by simple electroporation assays[15]. Additionally, the early *Ciona* embryo has exceptionally large, spherical nuclei (~8–10 microns in diameter) that undergo rapid mitotic divisions every 30–60 min, thereby facilitating the visualization of condensates throughout the cell cycle. In this study we used the *Sox1/2/3* enhancer to drive expression of different transcription factors in ectodermal cells, beginning at the onset of gastrulation at the 110-cell stage (Fig. 1a)[16]. These cells are nicely suited for the analysis of intracellular dynamics as they do not undergo the complex movements seen for the presumptive endoderm and mesoderm located on the other (vegetal) side of gastrulating embryos[17]. Coding sequences of interest were placed downstream of the *Sox1/2/3* enhancer, and fluorescent moieties such as mNeongreen (mNg, 236 amino acid residues) were fused in-frame at either the 5′ or 3′ position.

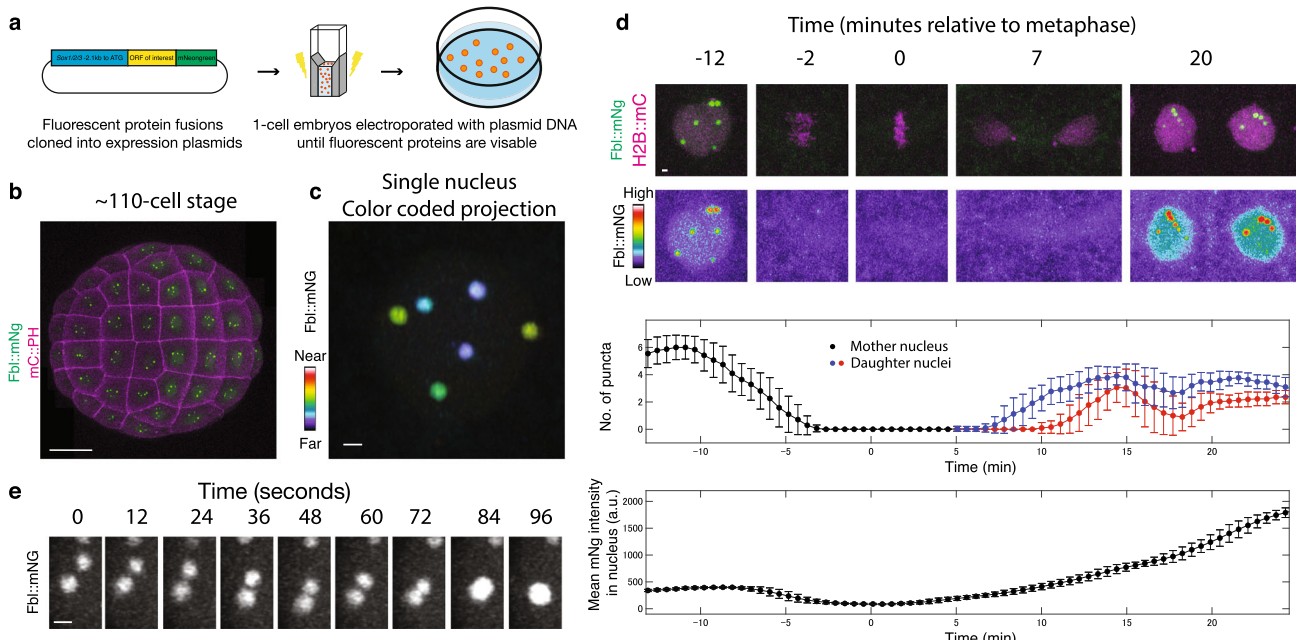

**Fig. 1 The nucleoli of *Ciona* embryos show liquid properties. a** Schematic of the electroporation procedure used to transfect *Ciona* embryos with plasmid DNA. **b** Confocal maximum intensity projection of a whole *Ciona* embryo expressing a Fbl::mNg transgene Scale bar = 20 μm. **c** Color coded projections of a single nucleus expressing Fbl::mNg. Scale bar = 1 μm. **d** Time-lapse maximum intensity projection confocal images of a single *Ciona* nucleus from the 7th to 8th mitosis. Fbl::mNg is shown in green and with the indicated look up table. H2B::mCh is shown in magenta. Graphs depict properties of green fluorescence within the red fluorescence region. Scale bar = 1 μm. Each central data point is the average over 30 frames (see methods). Error bars show the standard deviation. Scale bar = 1 μm. **e** Time-lapse maximum intensity projection confocal images of the fusion of 2 nucleoli. Scale bar = 0.5 μm. All images are representative of >3 biological replicates. Source data are provided as a Source Data file.

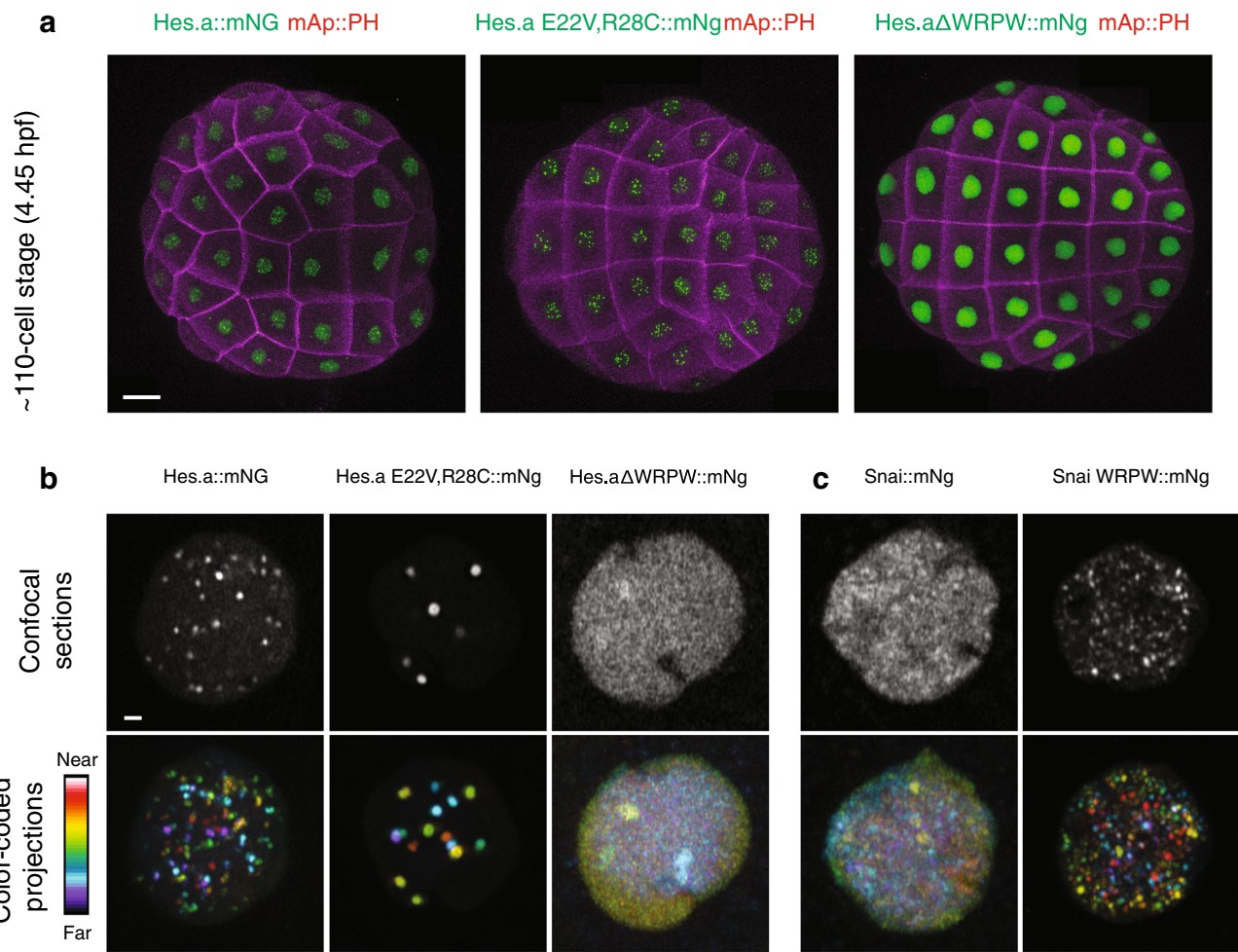

**Fig. 2 The Hes.a repressor forms puncta in *Ciona* embryos dependent on DNA binding and the presence of a WRPW domain. a** Maximum intensity confocal projections of ~110-cell stage embryos expressing Hes.a fusion proteins. Cell membranes are colored magenta and Hes.a::mNg fusion proteins are green. The embryos are oriented to show the animal hemisphere, anterior left. Scale bar = 20 μm. **b** Confocal images of individual nuclei expressing Hes.a proteins fused to mNg. Single confocal sections are shown in white, color coded projections are shown with the indicated look up table. **c** Same as **b** but for the Snai::mNg. Scale bar = 1 μm. All transgenes were expressed using pSP *Sox1/2/3 >* plasmids. All images are representative of >3 biological replicates.

As proof of principle, we examined the distribution of the Fibrillarin (Fbl) protein, an integral component of the nucleolus[18] (Fig. 1b). The *Ciona* genome contains three clusters of rRNA genes[19], and as expected for diploid embryos, we observe up to six distinct nucleoli of variable sizes (Fig. 1b, c). This assay captures dynamic dissolution and reassembly of individual nucleoli during the cell cycle (Fig. 1d, Supplementary Movie 1), which is likely to be a shared property of other nuclear condensates[20]. As shown for nucleoli in *Xenopus* oocytes[3], *Ciona* nucleoli display properties of viscous liquid droplets that undergo variable fusions (Fig. 1e, Supplementary Movie 2). We therefore conclude that the *Ciona* assay is nicely suited for visualizing nuclear condensates in living embryos, including dynamic droplet-droplet fusions.

**Hes.a repression puncta.** We employed this assay to examine transcription factors that are known to be important for early *Ciona* development, including Brachyury, Foxa.a, and Tfap2[21] (Supplementary Figure 1). Each displays distinctive sub-nuclear distribution profiles. Strikingly, we found that Hes repressor proteins, such as Hes.a, are distributed in spherical puncta in regions of the *Ciona* genome that are likely to contain clusters of Hes.a binding sites (Fig. 2a, b). We investigated the properties of these puncta to see how closely they resemble liquid–liquid phase separated condensates, as seen for nucleoli (Fig. 1). Particular

efforts focused on two different Hes.a protein variants. The first contains two amino acid substitutions (E22V and R28C) in the bHLH domain that eliminate DNA binding[22]. The other lacks the WRPW peptide motif at the C-terminus that is essential for interactions with Gro. Both variants are inactive in vivo and lack the capacity to repress gene expression[11,22].

The loss of DNA binding leads to the formation of large Hes.a puncta, whereas loss of interactions with Gro causes the opposite phenotype—virtual elimination of puncta (Fig. 2a, b). The deletion of the WRPW motif had similar effects on the nuclear distribution of Hes.b and Hes.c (Supplementary Fig. 1). Remarkably, the WRPW peptide motif is sufficient to confer clustering of heterotypic DNA binding proteins that normally display dispersed distribution profiles, such as Snail. Snail:WRPW fusion proteins are distributed in puncta that are similar to those seen for Hes.a (Fig. 2c). The addition of WRPW to other transcription factors produced similar results (Supplementary Fig. 1).

Because it is difficult to measure absolute expression levels in these electroporation experiments, we measured fluorescence levels in multiple embryos over a range of electroporation conditions. Hes.a puncta could be seen over a ~10-fold difference in Hes.a expression levels, including low levels of fluorescence that are barely detectable above background levels (Supplementary Fig. 2a). Conversely, Hes.a WRPW deletion mutants always

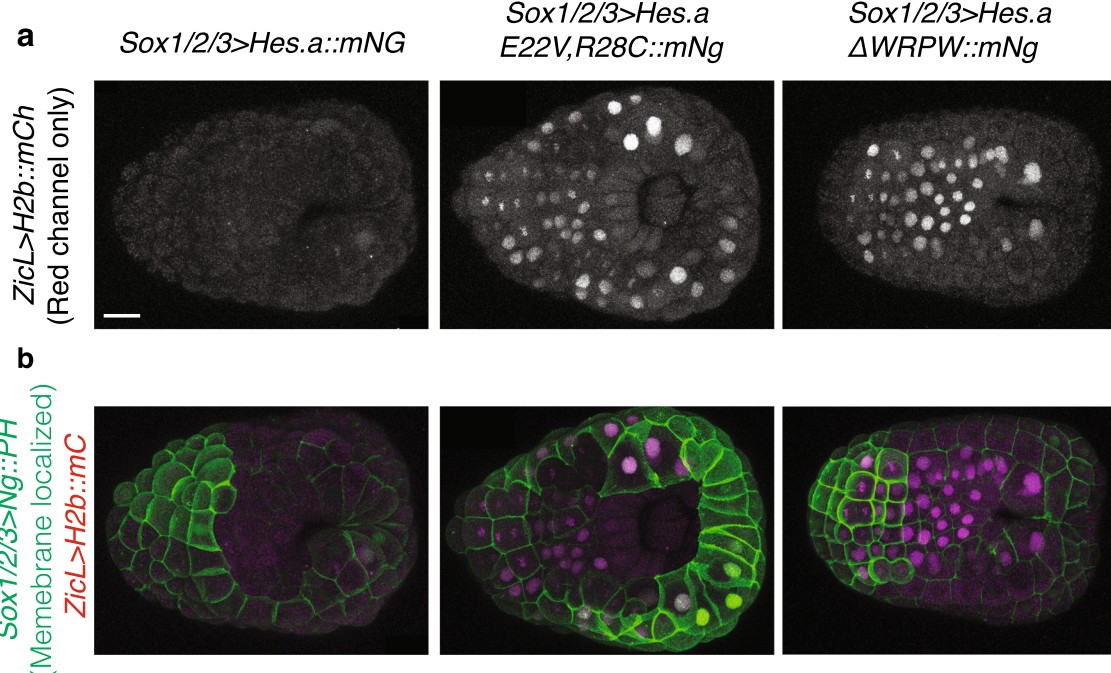

**Fig. 3 Hes.a represses *ZicL*. a** Red fluorescence in in gastrula stage *Ciona* embryos that have been from a *ZicL > H2b::mC* reporter plasmid. The embryos have been coelectroporated with Hes.a fusion proteins that were expressed earlier in development using plasmids containing the *Sox1/2/3* regulatory region. **b** Merged green and red fluorescent channel from the image in **a**. Because most of the Hes.a proteins have been degraded by this stage, cells that were previously expressing Hes.a transgenes were determined by the presence of the membrane localized mNG::PH fusion protein expressed using the *Sox1/2/3* regulatory region. Scale bar = 20 μm. All images are representative of >3 biological replicates.

displayed a uniform distribution, even at the highest levels (Supplementary Fig. 2b). While we cannot completely rule out any effects of overexpression, the presence of Hes.a-Gro puncta at even the lowest observable levels suggests that they are not an overexpression artefact. Further work on endogenous proteins will be required to definitively answer this question.

To determine whether Hes.a-Gro puncta correlate with transcriptional repression we examined the activities of a *ZicL > H2b::mCherry* (mCh) reporter gene. ZicL is an authentic target of the Hes.a repressor in early development[23]. Wild-type Hes.a efficiently represses the ZicL reporter, whereas mutant forms that are unable to bind DNA or interact with Gro do not (Fig. 3). These results show that the Hes.a::mNG fusion protein is capable of recapitulating the gene silencing activities of the native Hes.a repressor, and further suggest that Hes.a puncta are essential for repression.

**Other repression puncta.** Previous studies have shown that heterochromatin is compartmentalized within the nucleus. HP1 binds constitutive heterochromatin (H3K9me3)[24,25] and coalesces in living *Drosophila* embryos and cultured cells to form several large condensates located near the periphery of the nucleus[26,27]. Polycomb repression complexes bind to facultative heterochromatin (H3K27me3)[28] and also form higher order puncta resembling condensates[29]. Double labeling assays were used to determine whether Hes.a-Gro condensates are associated with either type of heterochromatin. Control experiments document co-localization of Hes.a and Gro (Fig. 4a), and loss of colocalization upon removal of WRPW from Hes.a (Supplementary Fig. 3a). By contrast, co-expression of the Hes.a::Ng with Fbl (nucleoli), Cdyl (a protein associated with PRC2 and H3K27me3[30]), or HP1 fusion proteins reveals little or no colocalization (Fig. 4; Supplementary Fig. 3b). These observations suggest that Hes.a does not silence gene expression by associating with heterochromatin, although it shares the property of forming puncta.

**Repression condensates.** We next examined the dynamics of Hes.a puncta to determine if they display liquid-like properties associated with LLPS. Both the wild-type (Fig. 5a, Supplementary Movie 3) and DNA binding mutant (E22V, R28C; Fig. 5b, Supplementary Movie 4) produce puncta that are detected throughout interphase, but are abolished during mitosis before reforming in daughter nuclei. The mutant exhibits faster dynamics than the wild-type protein. It dissolves more quickly during mitosis and re-forms more rapidly in daughter nuclei following mitosis. These results raise the possibility that the binding of Hes.a to its cognate DNA recognition sequences limits phase separation processes to specific nanoscopic regions of the genome (see Discussion).

Hes.a-Gro condensates appear to be more stable than activation condensates, which typically display short half-lives[5]. In contrast, Hes.a-Gro condensates are evocative of nucleoli, in that they display infrequent fusion events. Once formed, they persist throughout the cell cycle and do not dissolve until mitosis. Moreover, there is a progressive reduction in the number of wild-type Hes.a-Gro puncta during multiple cell cycles without a corresponding diminishment in fluorescence intensity. A possible explanation for this observation is that wild-type puncta undergo occasional fusion events as nuclei diminish in size, similar to fusions of individual nucleoli.

We next assessed Hes droplets for their ability to recover fluorescence after photobleaching (FRAP). Although FRAP is not a definitive property of liquid condensates it does suggest a dynamic composition that is consistent with LLPS[31]. Hes.a condensates exhibit recovery on the order of 15–30 s (Fig. 6a, Supplementary Movie 5), comparable to Pgl-1 subunits of P granules in *C. elegans*[32]. A faster rate of recovery was also seen for the Hes.a DNA binding mutant (Fig. 6a, Supplementary Movie 6).

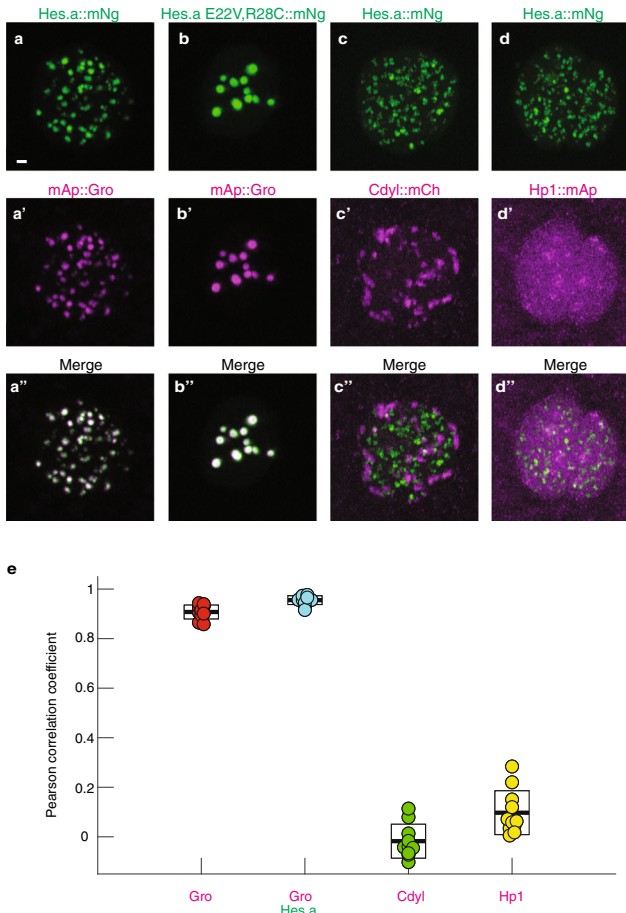

**Fig. 4 Hes.a/Groucho puncta are distinct molecular condensates. a** Maximum intensity projection confocal images of single *Ciona* nuclei expressing Hes.a::mNg. **a′** mAp::Gro Fluorescence. **a″** The merged green and red channels for **a** and **a′**. Scale bar = 1 μm. **b–b″**, As for the a series but for Hes.a E22V,R28C::mNG and mAp::Gro Green and red channels are shown individually and merged. **c–c″**, Hes.a::mNg and Cdyl:mCh d-d″, Hes.a::mNg and Hp1::mAp. **c** Pearson correlation coefficients of the experiments shown in **a–d″**. Boxes display the mean as the central line and the limits of the box denote standard deviations. All transgenes were expressed using pSP *Sox1/2/3* > plasmids. All images are representative of >3 biological replicates. Source data are provided as a Source Data file.

The high mobility as well as variable size and brightness of these condensates complicated quantitative measurements, but they were always found to exhibit definitive FRAP behaviors.

One of the defining properties of LLPS is the rapid fusions of individual condensates[31,33]. As discussed above, we can infer the occurrence of occasional fusions of Hes.a condensates during successive cell cycles (Fig. 6b). However, we were unable to detect these fusions without perturbations such as treatment with latrunculin A (Fig. 6b, Supplementary Movies 7, 8). In contrast, dynamic fusions were readily visualized for Hes.a DNA binding mutants (Fig. 6b, Supplementary Movie 9). Altogether, the preceding observations strongly point to the formation of Hes condensates by LLPS.

**Repression condensates in human cells**. We next asked whether Hes-Gro/TLE complexes form condensates in cultured HEK293 cells. The Human HES1 and TLE1 proteins displays a distribution profile that is similar to what we observed in the *Ciona* assay

(Supplementary Fig. 4), suggesting that this is a conserved property of Hes/Hairy repressors and associated Gro/TLE co-epressor proteins. To test the idea that oligomerization can drive phase separation of these proteins we utilized the recently developed corelet optogenetic system[34]. TLE corelets formed and were able to recruit HES1::GFP colocalized puncta upon light activation in cultured cells (Supplementary Fig. 4a; Supplementary Movie 10), and HES1 was also able to induce the formation of corelets (Supplementary Fig. 4b, Supplementary Movie 11). A HES1 DNA binding mutant (E43V,R49C) comparable to the *Ciona* Hes.a mutant (E22V,R28C) was found to produce droplets that are more dynamic than those seen for the normal Hes1 protein (Supplementary Fig. 4c, Supplementary Movie 12). These observations strengthen the evidence that Hes repressors form condensates by LLPS.

## Discussion

We have presented evidence that Hes.a-Gro complexes form condensates through LLPS. These condensates are likely to depend on dynamic interactions between the Hes.a and Gro subunits since puncta of either Hes or Gro do not form upon removal of the WRPW motif in Hes.a (Supplementary Fig. 3a). However, we could detect Gro puncta at later (tailbud) stages (Supplementary Fig. 5), suggesting the occurrence of authentic repressive Hes-Gro condensates. Gro proteins contain oligomerization and disordered domains in addition to WD40 repeats (Supplementary Fig. 6). These domains have been implicated in the formation of extended oligomers along the chromatin template[35]. There is emerging evidence that LLPS depends on the combination of oligomerization and disordered domains[34]. We suggest that such interactions trigger the formation of Hes.a-Gro condensates, similar to the formation of P granules and nucleoli[32,36]. However the growth and coarsening of these condensates appears to be limited by DNA binding since the E22V, R28C mutant displays dynamic fusion events that produce considerably larger droplets as compared with the wild-type protein.

Fusions of nucleoli have been shown to be suppressed by nuclear actin in *Xenopus* oocytes[37]. It is possible that actin limits chromatin diffusion and thereby limits fusions of wild-type Hes.a condensates. Another possibility is that these are secondary effects due to perturbing perinuclear actin. We suggest that the wild-type Hes repressor exhibits different properties upon binding to DNA. Recent work supporting this possibility comes from observations that condensate growth within a nucleus is limited by the surrounding chromatin[38], if a condensate is bound to DNA this effect could be amplified. Additionally, classical genetic studies have identified several examples of insulators delimiting the spreading of silencers into neighboring, active loci[13,39,40]. We therefore suggest that insulators serve to restrict the growth and fusions of repressive condensates, protecting actively transcribing regions (Fig. 7a).

How do Hes condensates mediate repression? One possibility is that they exclude transcriptional activators, coactivator complexes (e.g., Mediator), or active chromatin from inactive nuclear compartments, inverting the mechanism through which activation condensates recruit these proteins[41]. According to this view, silent and active regions of the genome form separate, immiscible droplets, analogous to oil and water. A nonexclusive possibility is that Hes condensates function as "repression factories" by recruiting histone deacetylases such as Rpd3[42], resulting in high localized concentrations of these enzymes (Fig. 7b). This scenario has been clearly described for photosynthetic condensates in algae—pyrenoids—which concentrate the carbon fixing enzyme rubisco[43]. A similar mechanism could also apply to heterochromatin since both HP1 and Polycomb form condensates that interact with histone methyltransferases[26,29].

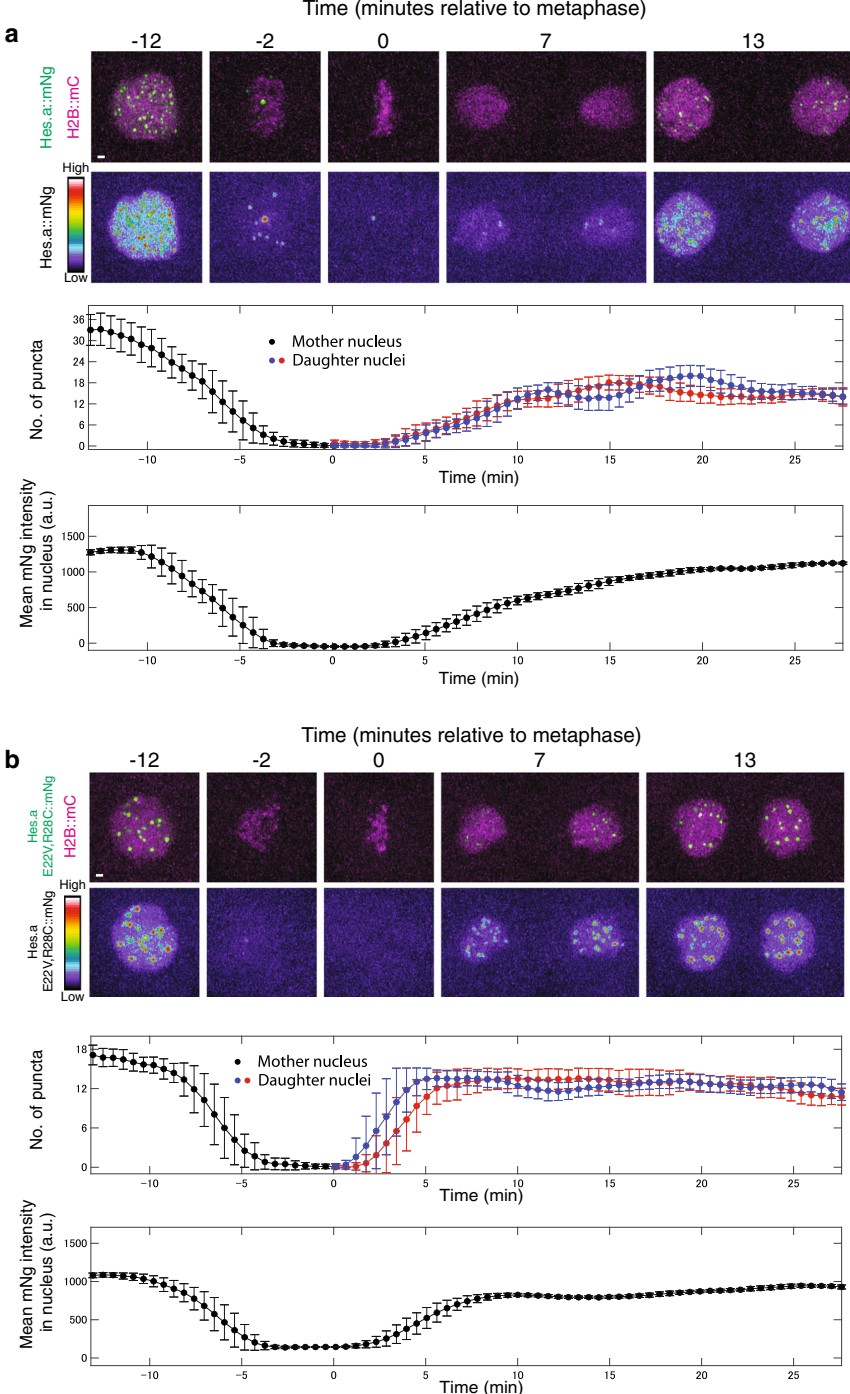

**Fig. 5 Hes.a shows liquid-like properties throughout the cell cycle. a** Time-lapse maximum intensity projection confocal images of a single *Ciona* nucleus from the 7th to 8th mitosis. Hes.a::mNg is shown in green and with the indicated look up table. H2B::mCh is shown in magenta. Graphs are depicting properties of green fluorescence within the red fluorescence region. Scale bar = 1 μm. Each central data point is the average over 30 frames (see methods). Error bars show the standard deviation. **b** Same as a but for the Hes.a E22V,R28C mutant. All transgenes were expressed using pSP *Sox1/2/3* > plasmids. All images are representative of >3 biological replicates. Source data are provided as a Source Data file.

## Methods

**Animals**. Wild type *Ciona intestinalis* (Type A, also recently referred to as *Ciona robusta*) sourced from San Diego County, Ca were supplied by M-Rep. All relevant ethical standards were complied with. Animals were kept in aerated artificial seawater at 18 °C. All procedures involving live animals were performed at ~18 °C. Latrunculin A treatments were done by replacing the artificial seawater with artificial seawater containing 0.1 μM Latrunculin A (Sigma) and immediately mounting and imaging the embryos.

**Human cells**. Corelet containing cells were all HEK293. Plasmids were introduced into cells by leniviral transfection using Lipofectamine 3000 and harvesting cells as previously described[34].

**Molecular cloning**. The upstream regulatory region of Sox1/2/3 from the translation initiation site to 2.3 kb upstream has previously been described to activate transgene expression in the ectoderm[16]. This sequence was subcloned into pSP

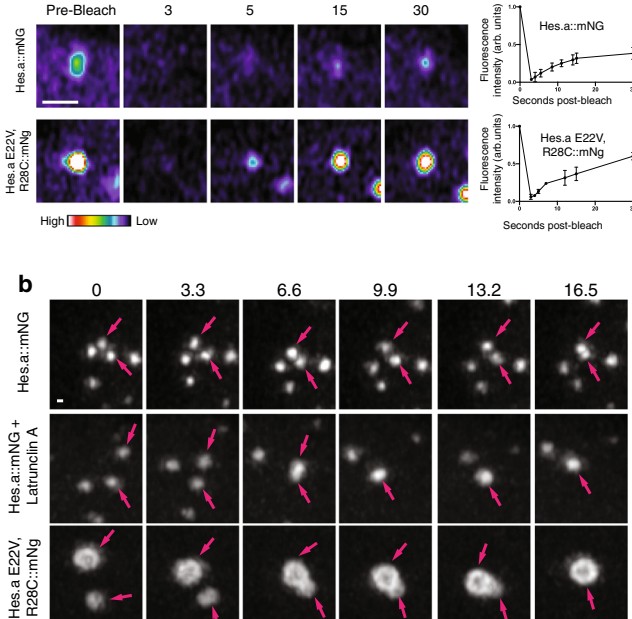

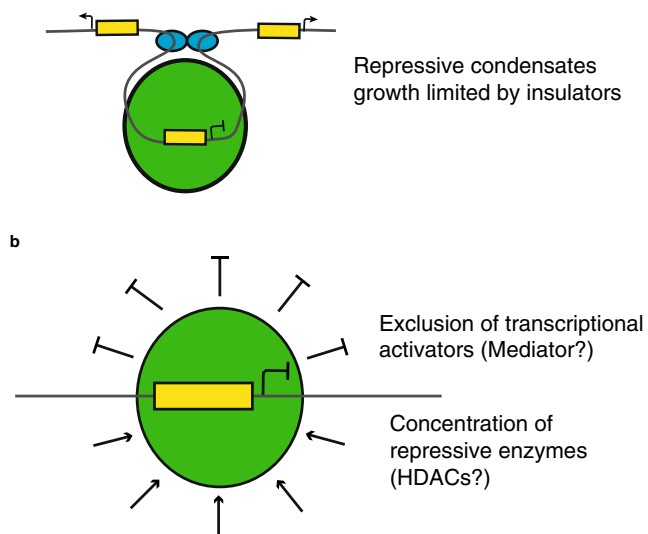

**Fig. 6 Hes.a FRAP/fusions. a** Time-lapse maximum intensity projection confocal images of region of a single Hes.a puncta. Images show the puncta prior to photobleaching and the recovery at several timepoints. Fluorescent intensity is indicated by the look up table. Scale bar = 1 μm. **b** Quantification of FRAP recovery of Hes.a droplets. Dots indicate the mean and the error bars indicate the standard deviation ($N = 3$). **c** Time-lapse maximum intensity projection confocal images of Hes.a puncta. 2 puncta are indicated with arrows and if they fuse are indicated by a single arrow. Scale bar = 0.5 μm. All transgenes were expressed using pSP Sox1/2/3 > plasmids. FRAP and fusion images are representative of >3 biological replicates. Source data are provided as a Source Data file.

**Fig. 7 The properties of repression condensates. a** This schematic depicts a Hes.a repression condensate (Green circle) bound to an enhancer (yellow block) DNA and silencing the appropriate gene. The growth of this condensate is limited by the presence of paired insulators (blue ovals). A consequence of this is that the repression condensate cannot spread to inappropriately silence the neighboring genes. **b** A proposed mechanism for how repression condensates can function based on the exclusion of transcriptional activators as well as creating a high concentration of enzymes known to be involved in transcriptional repression such as HDACs.

plasmids and the open reading frame of the gene of interest was amplified by PCR (see Supplementary Table 1) using a proofreading polymerase (Primestar, Takara). The open reading frames were fused to in frame to fluorescent protein coding sequences separated by the linker sequence: GGSGGGSGG. Plasmids were assembled from linear PCR products by treatment with NEBuilder HiFi DNA Assembly Master Mix (New England Biolabs). Full plasmid sequences and descriptions of the individual cloning steps can be provided upon request. The ZicL>H2B::mCherry plasmid has previously been described[44]. Corelet plasmids were constructed by ligating human HES1 and TLE cDNAs into pHR-SFFVp plasmids for lentiviral transfection as previously described[34]. Corelet plasmids were co-transfected with plasmids encoding nuclear localized iLIDs to permit light induced aggregation[34].

**Electroporation**. Dechorionated *Ciona* Zygotes were electroporated at 30 min post fertilization using standard electroporation settings[15]. Except where stated otherwise, 30 ug of plasmid DNA was electroporated for each individual plasmid used except for ZicL > H2B::mCherry where 20 ug was electroporated. For all reported experiments multiple replicate electroporations were performed using different batches of *Ciona* eggs. All described phenotypes were observed in several hundred nuclei and no meaningful embryo to embryo variation was seen.

**Imaging**. *Ciona* embryos were imaged using a Zeiss LSM 880 inverted confocal microscope (Carl Zeiss). Embryos were mounted on 3.5 cm glass bottom dishes (MatTek cat # P35G-1.5-20-C) as previously described[45]. Whole embryos were imaged using a 40 × 1.2 NA C-apochromat water immersion objective. Other images were taken with a 63 × 1.4 NA plan-apochromat oil immersion objective. All imaging was performed using an Airyscan detector in fast mode. FRAP was performed using Images were processed using ZEN software (ZEN Version 2.3 and 2.6, Zeiss). Human cell imaging was performed using a Nikon A1 laser scanning confocal microscope equipped with a $CO_2$ microscope stage incubator under 5% $CO_2$ and 37 °C with a plan-apochromat 60 × 1.4 NA oil immersion objective.

**Image analysis**. For all experiments involving electroporated *Ciona* embryos at least 3 biological replicates utilizing several hundred embryos were performed. Representative images were chosen for whole embryos that showed uniform expression of transgenes in all ectodermal cells. For individual nuclei representative images were chosen from observations of at least 300 nuclei in multiple embryos. Droplets fusions were observed in at least 3 different nuclei and FRAP was performed on at least 10 nuclei.

The colocalization between green and red fluorescent channels was quantified with pixel-based intensity correlation pearson correlation coefficients[41] where 1 is a perfect correlation, 0 is no correlation and -1 is perfect anti-correlation. For the time course calculations in Figs. 1 and 5, number of puncta and florescence intensity were calculated using custom Matlab scripts. Due to the noise of the data, averages and standard deviation were calculated for every 30 data points (6.6 s per frame). Error bars show the standard deviation. All custom code is available upon request.

**Prediction of disordered regions**. The prediction of disordered regions within Hes.a and Gro were determined from the translated open reading frames of predicted transcripts using the PONDR prediction software with the VSL2 prediction model[46].

**Reporting summary**. Further information on research design is available in the Nature Research Reporting Summary linked to this article.

## Data availability
All relevant data supporting the key findings of this study are available within the article and its Supplementary Information files or from the corresponding author upon reasonable request. A source data file is provided with this paper. A reporting summary for this Article is available as a Supplementary Information file. Source data are provided with this paper.

## Code availability
Custom MATLAB code used in this study is available upon request.

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

## Acknowledgements

We thank members of the Levine and Brangwynne labs for their support, especially Laurence Lemaire for sharing reagents and Evangelos Gatzogiannis for help with imaging. This research was funded by NIH grants (NS076542 to M.S.L.; 01 DA040601 to C.P.B.) and the HHMI (to C.P.B.). N.T. is funded by a Princeton Catalysis Initiative grant (to M.S.L. and C.P.B.). J.E. is funded by an NWO Rubicon grant.

## Author contributions

N.T. and M.S.L. conceived the project and designed the experiments. N.T. performed the *Ciona* experiments. S.F.S. performed image analysis. J.E. and C.P.B. designed and J.E. performed human cell experiments. All authors interpreted the data. N.T. and M.S.L. wrote the paper with input from all other authors.

## Competing interests

The authors declare no competing interests.
