## [Peer Review File · Nature Communications]

Reviewers' comments:

Reviewer #1 (Remarks to the Author):

Treen et al present evidence that the Hes.a and Gro proteins form condensates in Ciona embryos. They utilize an electroporation approach to introduce fluorescently labeled constructs into Ciona allowing them to visualize Hes/Gro condensates in live embryos. They find that DNA binding domain (DBD) mutants of Hes.a yield larger and fewer condensates, likely due to coalescence of smaller condensates no longer tethered to specific genomic loci. They find that the WRPW domain of Hes.a is required for condensate formation and can confer condensate formation capacity onto other proteins. Treen et al report that the Hes.a/Gro condensates dissolve and reform through mitosis, a feature that has been observed for the majority of nuclear condensates studied so far. Lastly, through co-localization studies Treen et al shows that while Gro and Hes.a colocalize within the Hes.a/Gro condensates they do not co-localize with CDYL or HP1. Overall the manuscript is preliminary and not ready for publication in Nature Communications. This study is perhaps better suited for a “brief communication” or “short report” format offered by other journals, but I would recommend additional experiments and analysis even for those formats.

Major comments:

1. Given the well-documented concentration dependence of phase separation, the lack of systematic analysis of protein expression levels is problematic. It is important to know both the expression relative to endogenous protein, and the relative expression levels of the different electroporated constructs being compared. The RT-PCR data presented in Figure S1 is insufficient. The data from S1, show that there is a ~20-fold overexpression of Hes.a from a population of hundreds of pooled embryos. How large is the distribution of expression across populations of embryos? Is this the same with each construct? Is this the same every time an electroporation is performed?
2. All images are presented as representative without any population analysis (within an embryo or across many embryos). In particular, given the unknown but likely high distribution of expression levels across the electroporated population (see comment 1), results may vary by expression level and should be reported. Representative images should be accompanied by a population analysis. It is difficult to interpret the results without this type of information.
3. The authors make a point about highlighting the WD40 repeats as a potential general domain responsible for multivalent interactions leading to condensate formation. This is an interesting idea, but is not supported by data in the manuscript. At a minimum, the authors should use a construct without WD40 repeats to test their contribution to condensate formation. The contribution of the WRPW domain is interesting, but it is indirect.
4. It appears that wildtype Hes.a forms cytoplasmic puncta that are not present in the DBD mutant. This is striking in the representative images presented in figure 1. The authors do not comment on this. Does Hes.a have a known cytoplasmic function?
5. In the images presented for the functional reporter assays, it appears as though Hes.a is not in the nucleus. The staining looks more like the magenta membrane signal in figure 1 than the Hes.a signal. Is this a mistake in the data presentation and/or figure legend? If this is not a mistake, then it is confusing to understand how the Hes.a condensates are responsible for silencing when there is no nuclear Hes.a signal let alone Hes.a puncta present?
6. The authors are suggesting that Hes.a/Gro condensates are responsible for silencing by forming locus-tethered condensates (this is the model presented in figure 4), but there is no evidence presented that these condensates are present at repressed loci. The authors need to perform immunofluorescence coupled with DNA-FISH for classic Hes/Gro targets to show that these puncta are present at those

locations or comparable experiments.

7. The title is not supported by the data in the paper. The link between “Regulation of Gene Expression” and the observed “repression condensates” is not made by data in this paper.

Minor comments:

1. A diagram of the constructs used would be helpful to the reader.
2. “neither protein alone forms puncta” Where is this demonstrated?
3. “Gro has 7 WD40 motifs...” This is confusing. Where has it been shown that the 7 WD40 motifs are required for formation of condensates? That data is not present in this manuscript and not found in the reference cited. Is this a typo? Should it read that the 7 WD40 motifs are required for Gro’s repression activity?

Reviewer #2 (Remarks to the Author):

This manuscript addresses a problem how the Hes/Gro repressor complex represses gene expression, and presents evidence that this repression is achieved by formation of compartments assembled through liquid-liquid phase separation (LLPS). Although LLPS has been suggested to explain the formation and regulation of superenhancers, it has not been suggested to be associated with transcriptional repression. The authors provides evidence that the Hes/Gro repressor complex represses gene expression through LLPS. The finding that Hes/Gro condensates are relatively long lived is interesting and potentially important.

I have several minor points. I believe that the authors can easily fix these points.

- 1 This manuscript was probably written for a different journal. But the authors can use more words to explain details in this journal. I strongly suggest the authors to include more details in the manuscript.
 - 1.1 Ext Fig.1 shows the expression levels of two different constructs, but the main text describes only Sox1>Sox1::mNg. Describe the second one, which is more important than the first one.
 - 1.2 Ext. Fig1 b : it is not clearly described how they measured the expression levels of these two constructs.
 - 1.3 The legend says that error bars in Ext. Fig1b represent standard errors, but for what? Technical or biological replicates? How many replicates did they take?
 - 1.4 Similarly, details for Ext Figures 2, 3, 5, and 7 should be provided in the main text and legends.
 - 1.5 Some ext figs could be presented as main ones.
- 2 For Fig 1b, c, Fig 2, ExtFig2, ExtFig3, ExtFig4, ExtFig7, how many embryos/cells/nuclei did they examine? How many independent experiments (independent electroporation/transfection experiments) did they do? It should be described clearly in the manuscript.
- 3 Similarly, for Fig. 3 and Ext Fig5, how many independent experiments (independent electroporation/transfection experiments) did they do?
- 4 Line 105: The authors report 'a progressive reduction in the number of wild-type Hes.a/Gro puncta'. The authors say this occurs during multiple cell cycles, but they observed only one cell cycle. As a result, the claim described in the next sentence becomes less persuasive.
- 5 Fig.2 legend: I was not able to understand the sentence 'Error bars show the standard deviation +- 100 sec'. Indicate what black, blue and red dots mean, too.

6 Line 114-122, Fig 3d: Localization of HP1 is not clear enough. It may be possible that this protein was not localized in heterochromatin. It will be a good idea to replace this photograph.

7 Line 151: The authors say that seven WD40 repeats of Gro are required for the formation of condensates. But this is not examined. It will be easy to examine this possibility by taking advantages of the experimental system used in this study. It will strengthen the authors' claim. If the authors do not want to do this experiment, the tone of this paragraph should be softened.

Reviewer #3 (Remarks to the Author):

In this manuscript Treen et al. investigate whether transcriptional repression also involves the formation of transcription condensates, as has recently been shown for transcriptional activation. For this they explore the role of liquid-liquid phase separation (LLPS) for the Groucho/TLE (Gro) family of transcriptional corepressors and their interaction with sequence specific repressors from the Hes/Hairy family. They express fluorescent fusion proteins of interest in *Drosophila* embryos, allowing them to visualize their (co)localization and dynamics in living cells.

Using this assay, they show that Hes/Gro complexes form discrete puncta within nuclei of living *Drosophila* embryos. By expressing Hes.a mutants they show that these puncta become larger when Hes.a can no longer bind DNA and are absent when Hes.a loses its WRPW Gro interaction domain. Using a reporter gene, they show that only wild type Hes.a is able to repress, whereas the mutants unable to bind DNA or Gro do not.

Looking at the dynamics of Hes/Gro puncta, they make the observation that Hes/Gro puncta are long lived, when compared to the short half-lives previously observed for activator condensates. They also show that these puncta rapidly dissolve during the onset of mitosis and reappear in the following cell cycle. Using the Hes.a DNA binding mutant, they could observe fusion of condensates, which is considered a critical property of liquid-liquid phase separation. However, such fusion events were not observed when wild-type Hes.a proteins were expressed. Furthermore, they show that these Hes/Gro condensates do not co-localize with heterochromatin or nucleoli, suggesting that they are distinct and therefore silence genes through different means. Finally, making use of the corelet optogenetic system, they show that also in human cells Hes1 and TLE(Gro) can form colocalized puncta, be it upon forced nucleation through light activation.

Overall, this study makes several interesting observations concerning the potential role of liquid-liquid phase separation during transcriptional repression, which, as the authors point out, is an important yet understudied aspect of gene regulation. While droplet formation has been observed for Polycomb and HP1-associated heterochromatic regions, this study adds repressing transcription factors and the corepressors they recruit to the spectrum of repressive protein complex forming LLPS condensates. There are however several major concerns that need to be resolved prior to publication:

Major concerns

In the abstract and at the end of the manuscript the authors make several strong propositions and even generate a main figure (Fig.4) portraying how they think the repression model might work, without providing any actual data to directly support this (e.g. “We propose that repression condensates inhibit gene expression by the mechanical exclusion of transcriptional activators, coactivator complexes such as Mediator, or active chromatin (Fig. 4)”). Even though this model of activator exclusion due to repression droplet formation is attractive, the authors do not provide any data to directly support the

exclusion of activators from these droplets. The authors should derive conclusions and main figures from the data presented and perhaps move such speculations and propositions to a distinct discussion section (not present in the current version of the manuscript).

Their methods section does not provide the sufficient information to assess some of experiments done in this manuscript or to potentially reproduce it. E.g what has actually been done in the Human cell system is unclear and plasmid maps do not seem to be available. Did the authors use iLID domains induce the interactions and nucleation upon light stimulus or not? If they did, what does this mean for their model? The referrals to other papers don't really help and rather make this part unnecessarily difficult to assess and vague.

The authors state "There is only a ~3-fold increase in the levels of expression as compared with endogenous Sox 1/2/3 products (Extended Data Fig. 1)." While it is true that there is only a 3-fold increase compared to the endogenous Sox 1/2/3 products, the relevant comparison is with the endogenous levels of Hes.a, which suggests a ~20 fold increase. This needs to be explicit as it could be the source of artifacts (see below).

The authors should clarify, whether they have evidence that the diffuse green fluorescent signal derived from the mutant Hes.a_deltaWRPW in Fig. 1B, which completely fills out the whole nucleus, truly reflects a lack of puncta formation in the case of this mutant protein, or might be due to a very strong signal, making it harder to discern individual spots of aggregated protein.

In Extended Data Figure 4 the authors suggest that repression is dependent on the formation of Hes.a/Gro puncta. "These results suggest that the formation of the Hes.a puncta, binding to both DNA and Gro, are required for repression." They indeed show that for repression both Hes.a binding to DNA and Gro are required, which is known. They did however not show the presence of these puncta near repressed genes nor a causal requirement of formation of puncta for repression to occur. Hence, the relationship between the formation of repressor droplets and transcriptional repression is only correlational and fairly weak given that DNA binding and Gro recruitment are known to be necessary for repression. It should be made more obvious to the reader that such observations constitute mere coincidence rather than evidence suggestive of specific mechanisms.

Fig.3 appears as if Hes.a is the only factor that binds Gro, because Gro is only present in droplets that contain Hes.a. Yet, as for example shown in Extended Data Figure 3, also Hes.b and Hes.c have WRPW domains for the interaction with Groucho and form liquid-like condensates, as should many other transcription factors. The fact that we only see Hes.a puncta in Figure 3 but no evidence of any puncta for any of the other factors that must be present suggests that the Hes.a puncta are artefacts due to the artificially high (~20 fold up) expression levels of Hes.a (see above).

Minor remarks:

The legend of Fig. 1 states that puncta formation depends also on DNA binding of Hes.a, even though abolishing DNA binding leads to even larger puncta, as mentioned in the main text.

Fig. 2 should mention in the legend what blue and red colored data points refer to (We can only guess that they represent data from the two daughter cells).

"Such fusions are readily detected for the E22V,R28C mutant (Fig. 2C, Supplementary Video 5), but not for the wild-type protein. However, there is a progressive reduction in the number of wild-type Hes.a/Gro puncta during multiple cell cycles without a corresponding diminishment in fluorescence

intensity (Fig. 2A). A possible explanation for this observation is that wild-type puncta undergo fusion events as nuclei diminish in size, creating higher concentrations of compact chromatin as compared with earlier stages of development.”

Their explanation that wild-type puncta might undergo fusion events as nuclei diminish in size is rather strange, as they showed before that puncta are abolished and reformed during each cell division: “Both the wild-type and DNA binding mutant (E22V,R28C) produce puncta that are detected throughout interphase, but are abolished during mitosis before reforming in daughter nuclei (Fig. 2A, B Supplementary Videos 3, 4).”

The inability of repressor condensates to fuse when wt Hes.a is present, can be explained by immobilization of the droplets due to binding of target DNA. In our opinion this does not weaken the argument of LLPS being involved in transcriptional repression.

“We propose that interactions between proteins containing disordered domains with those containing WD40 repeats might be a key trigger for the oligomerization of biological condensates.” Please move such speculations that are not supported by experiments to a clearly separated discussion section and clearly state that these are hypotheses.

Please make sure the figure labels are clear and readable, e.g. extended data Fig.3,

Extended data Fig.4: why does it seem that Hes.a::mNG signal is mainly accumulated at the cell membranes, even for the wild type Hes.a protein? Why is the membrane protein and Hes.a protein localization shown in the same color (=green)? This makes it difficult to discern where Hes.a is actually localized.

Reviewer #1 (Remarks to the Author):

Treen et al present evidence that the Hes.a and Gro proteins form condensates in Ciona embryos. They utilize an electroporation approach to introduce fluorescently labeled constructs into Ciona allowing them to visualize Hes/Gro condensates in live embryos. They find that DNA binding domain (DBD) mutants of Hes.a yield larger and fewer condensates, likely due to coalescence of smaller condensates no longer tethered to specific genomic loci. They find that the WRPW domain of Hes.a is required for condensate formation and can confer condensate formation capacity onto other proteins. Treen et al report that the Hes.a/Gro condensates dissolve and reform through mitosis, a feature that has been observed for the majority of nuclear condensates studied so far. Lastly, through co-localization studies Treen et al shows that while Gro and Hes.a colocalize within the Hes.a/Gro condensates they do not co-localize with CDYL or HP1. Overall the manuscript is preliminary and not ready for publication in Nature Communications. This study is perhaps better suited for a “brief communication” or “short report” format offered by other journals, but I would recommend additional experiments and analysis even for those formats.

Overall Response: Obviously we disagree with the referee’s bottom line conclusion regarding the suitability of our paper for Nat Comm. We believe that our study deserves a bit more credit for the clarity of the imaging, the superior quality of the Ciona expression assay, and the fact that this is the very first analysis of a sequence-specific repressor. Groucho/TLE is a major corepressor protein that is highly conserved in evolution, from yeast to humans. Any new insights into underlying mechanism should be of interest to general readers of Nature Communications. In particular, we highlight a previously unappreciated role of the DNA template, namely, inhibition of LLPS. The switch in Hes.a behavior from large dynamic droplets (off DNA) to viscous gels (on DNA) is supported by newly added FRAP assays and drug inhibition experiments..

Below we provide a point-by-point rebuttal:

Major comments:

1. Given the well-documented concentration dependence of phase separation, the lack of systematic analysis of protein expression levels is problematic.

Response 1.1: We already addressed this question using qPCR assays to show that Hes.a expression levels are just a few-fold higher using Sox1/2/3 transgenes as compared with endogenous levels. More importantly, we show that Hes.a condensates are eliminated by removal of the simple WRPW motif at the C-terminus of the proteins. Strikingly, heterologous proteins, such as Snail, are shown to form condensates only upon addition of the WRPW motif. We believe that these observations provide compelling evidence that Hes.a-Gro complexes form condensates. We wish to emphasize the point that Hes.a condensates are always observed, even with a large range of electroporation conditions producing a broad spectrum of protein

concentrations. In contrast, we never see condensates with the WRPW mutant protein, regardless of expression levels.

It is important to know both the expression relative to endogenous protein, and the relative expression levels of the different electroporated constructs being compared. The RT-PCR data presented in Figure S1 is insufficient. The data from S1, show that there is a ~20-fold overexpression of Hes.a from a population of hundreds of pooled embryos. How large is the distribution of expression across populations of embryos?

Response 1.2: We have removed the RT-PCR data from the revised version since it is clearly misleading our readers. As we have already deposited these results in Biorxiv it will still be available for anyone interested. We included this data in order to provide a ballpark figure of overexpression levels. The 20-fold increase that the referee describes is mainly due to the larger number of cells (at least 4-5 fold more than normal) that now express Hes.a using the Sox1/2/3 transgene. That is why we considered comparison with endogenous Sox1/2/3 levels to be a more meaningful comparison.

Is this the same with each construct?

Response 1.3: We assume that this will be the same for each construct as they all use the same regulatory region (Sox1/2/3).

Is this the same every time an electroporation is performed?

Response 1.4: The RT-PCR experiments were performed in biological triplicates and the variability can be inferred from the error bars.

All images are presented as representative without any population analysis (within an embryo or across many embryos). In particular, given the likely variations in expression levels across populations of electroporated embryos (see comment 1), results may vary by expression level and should be reported. Representative images should be accompanied by a population analysis. It is difficult to interpret the results without this type of information.

Response to 1.5: We feel that population analyses would not be profitable given the variation and mosaicism that is a well-known property of this method. Transgenesis by electroporation has been used in the Ciona community since 1996; it is not a newfangled technology. It has been used very successfully to create mutant phenotypes, including the analysis of cell fate transformations using single-cell RNA sequencing assays. The key point of the images is to emphasize that we are performing these experiments in living embryos rather than cultured cells, or worse, test tubes. We are not presenting just a few cherry-picked snapshots, but rather, are attempting to accurately depict the events seen in hundreds of embryos and thousands of

nuclei. We have revised the Methods section to more directly state the number of replicates we have performed and how we settled on representative images.

The authors make a point about highlighting the WD40 repeats as a potential general domain responsible for multivalent interactions leading to condensate formation. This is an interesting idea, but is not supported by data in the manuscript. At a minimum, the authors should use a construct without WD40 repeats to test their contribution to condensate formation. The contribution of the WRPW domain is interesting, but it is indirect.

Response to 1.6: We agree with this comment and have toned down our emphasis on the WD40 repeats in the revised version. We have made many attempts to disrupt the WD40 domain of Grouco, but in almost all cases they fail to show nuclear localization and appear to be highly unstable, as reported in previous studies (Turki-Judeh and Courey 2012, Plos One).

It appears that wildtype Hes.a forms cytoplasmic puncta that are not present in the DBD mutant. This is striking in the representative images presented in figure 1. The authors do not comment on this. Does Hes.a have a known cytoplasmic function?

Response to 1.7: Hes.a does not have a known cytoplasmic function to our knowledge. The cytoplasmic Hes.a seen in the original figure 1 is not something seen in all cases, but it is not uncommon. We have replaced this image with another embryo that has less cytoplasmic puncta. However, the reviewer's suspicion is correct. This does not appear in the DNA binding mutant in any of our replicates. We do not have an explanation for this phenomenon.

In the images presented for the functional reporter assays, it appears as though Hes.a is not in the nucleus. The staining looks more like the magenta membrane signal in figure 1 than the Hes.a signal. Is this a mistake in the data presentation and/or figure legend? If this is not a mistake, then it is confusing to understand how the Hes.a condensates are responsible for silencing when there is no nuclear Hes.a signal let alone Hes.a puncta present?

Response to 1.8: We are aware that our presentation of the reporter assays can be confusing. We have modified the figure and legend to try and make it clearer. As we said in a previous comment, *Ciona* electroporations can be mosaic. So, if we see an absence of reporter signal this could be because the cell does not have the plasmid, rather than having the plasmid and it being repressed. Another complication is that Hes.a is quickly degraded. By the time we can see a reporter signal, Hes.a protein will mostly be gone. Our solution to this was to co-electroporate the Hes.a::Ng and ZicL>mCherry reporters with a membrane localized form of Ng(Ng::PH) This is highly stable and persists after the Hes.a signal is depleted. So, when we see membrane signals we know they arise from a cell that previously expressed Hes.a. We accept that this is confusing, and have tried to improve the description of this experiment in the figure legend.

The authors are suggesting that Hes.a/Gro condensates are responsible for silencing by forming locus-tethered condensates (this is the model presented in figure 4), but there is no evidence

presented that these condensates are present at repressed loci. The authors need to perform immunofluorescence coupled with DNA-FISH for classic Hes/Gro targets to show that these puncta are present at those locations or comparable experiments.

Response to 1.9: The main difficulty with this request is that the Hes.a puncta do not maintain their structure throughout fixation (because they are liquids!) particularly with the harsh chemicals and conditions required for FISH. We are aware that this would be an extremely useful experiment and we are attempting to find ways to directly visualize repression *in situ* in future studies.

The title is not supported by the data in the paper. The link between “Regulation of Gene Expression” and the observed “repression condensates” is not made by data in this paper.

Response to 1.10: We have changed the title.

Minor comments:

1. A diagram of the constructs used would be helpful to the reader.

Response 1.11: We are reluctant to do this because it would clutter the figures. Also, the constructs used are probably simpler than the reviewer assumes. With the exception of our ZicL>mCherry reporter, all our electroporation constructs follow the basic arrangement shown in Figure 1 with the Sox1/2/3 regulatory region and the fusion protein of interest. We have revised the figure legends to make this point clearer.

“neither protein alone forms puncta” Where is this demonstrated?

Response 1.12: We have modified this statement to more directly support our experiments. However, neither the WRPW deletion of Hes.a nor Groucho alone forms puncta when expressed as fluorescent fusion proteins so we believe this is true.

“Gro has 7 WD40 motifs...” This is confusing. Where has it been shown that the 7 WD40 motifs are required for formation of condensates? That data is not present in this manuscript and not found in the reference cited. Is this a typo? Should it read that the 7 WD40 motifs are required for Gro’s repression activity?

Response 1.13: This argument has been removed from the revised version.

Reviewer #2 (Remarks to the Author):

This manuscript addresses a problem how the Hes/Gro repressor complex represses gene expression, and presents evidence that this repression is achieved by formation of compartments assembled through liquid-liquid phase separation (LLPS). Although LLPS has been suggested to

explain the formation and regulation of superenhancers, it has not been suggested to be associated with transcriptional repression. The authors provide evidence that the Hes/Gro repressor complex represses gene expression through LLPS. The finding that Hes/Gro condensates are relatively long lived is interesting and potentially important.

I have several minor points. I believe that the authors can easily fix these points.

This manuscript was probably written for a different journal. But the authors can use more words to explain details in this journal. I strongly suggest the authors to include more details in the manuscript.

Response 2.1: The authors speculation is correct. This paper was originally submitted to Nature and was transferred to Nature Communications. We have expanded the paper to take advantage of the longer article size permitted in this journal. We have also extensively rewritten the paper in order to make the key points clearer.

Ext Fig.1 shows the expression levels of two different constructs, but the main text describes only Sox1>Sox1::mNg. Describe the second one, which is more important than the first one.

Response 2.2: This is covered in Referee 1 above, Response 1.1 and 1.2.

Ext. Fig1 b : it is not clearly described how they measured the expression levels of these two constructs.

Response 2.3: The construct expression levels were measured by qPCR from RNA treated to remove genomic and plasmid DNA using the primers described in Methods.

The legend says that error bars in Ext. Fig1b represent standard errors, but for what? Technical or biological replicates? How many replicates did they take?

Response 2.4: 3 biological replicates

Similarly, details for Ext Figures 2, 3, 5, and 7 should be provided in the main text and legends. Some ext figs could be presented as main ones.

Response 2.5: In the revised version some of the supplementary figures have been moved to the main text and we have provided more details in the legends and main text.

For Fig 1b, c, Fig 2, ExtFig2, ExtFig3, ExtFig4, ExtFig7, how many embryos/cells/nuclei did they examine? How many independent experiments (independent electroporation/transfection experiments) did they do? It should be described clearly in the manuscript.

Response 2.6: We have generally covered this in our response to reviewer 1 and the revised manuscript provide more details on the number of replicates performed.

Similarly, for Fig. 3 and Ext Fig5, how many independent experiments (independent electroporation/transfection experiments) did they do?

Response 2.7: Three separate transfections; several nuclei examined in each experiment.

Line 105: The authors report 'a progressive reduction in the number of wild-type Hes.a/Gro puncta'. The authors say this occurs during multiple cell cycles, but they observed only one cell cycle. As a result, the claim described in the next sentence becomes less persuasive.

Response to 2.8: We have revised the text to emphasize that this interpretation is purely speculative.

Fig.2 legend: I was not able to understand the sentence 'Error bars show the standard deviation +/-100 sec'. Indicate what black, blue and red dots mean, too.

Response 2.8: The meaning of the black, blue and red dots is now indicated on the graph. We apologize for omitting this from the initial submission. We calculated the error bars for both the x and y axis because the data has approximately 30 data points for every 100 seconds so we felt this was the best way to represent these measurements.

Line 114-122, Fig 3d: Localization of HP1 is not clear enough. It may be possible that this protein was not localized in heterochromatin. It will be a good idea to replace this photograph.

Response 2.9: We agree that Hp1:mApple is not clear but we could not get a better red fusion for this protein. The distribution is similar to the clearer Hp1::mNG so we are confident this is localizing to heterochromatin.

Line 151: The authors say that seven WD40 repeats of Gro are required for the formation of condensates. But this is not examined. It will be easy to examine this possibility by taking advantages of the experimental system used in this study. It will strengthen the authors' claim. If the authors do not want to do this experiment, the tone of this paragraph should be softened.

Response 2.10: We have covered this in our response to reviewer 1. We have toned down our discussion on the WD40 repeats.

Reviewer #3 (Remarks to the Author):

In this manuscript Treen et al. investigate whether transcriptional repression also involves the formation of transcription condensates, as has recently been shown for transcriptional activation. For this they explore the role of liquid-liquid phase separation (LLPS) for the Groucho/TLE (Gro) family of transcriptional corepressors and their interaction with sequence specific repressors from the Hes/Hairy family. They express fluorescent fusion proteins of interest in Ciona embryos, allowing them to visualize their (co)localization and dynamics in living cells.

Using this assay, they show that Hes/Gro complexes form discrete puncta within nuclei of living Ciona embryos. By expressing Hes.a mutants they show that these puncta become larger when Hes.a can no longer bind DNA and are absent when Hes.a loses its WRPW Gro interaction domain. Using a reporter gene, they show that only wild type Hes.a is able to repress, whereas the mutants unable to bind DNA or Gro do not.

Looking at the dynamics of Hes/Gro puncta, they make the observation that Hes/Gro puncta are long lived, when compared to the short half-lives previously observed for activator condensates. They also show that these puncta rapidly dissolve during the onset of mitosis and reappear in the following cell cycle. Using the Hes.a DNA binding mutant, they could observe fusion of condensates, which is considered a critical property of liquid-liquid phase separation. However, such fusion events were not observed when wild-type Hes.a proteins were expressed. Furthermore, they show that these Hes/Gro condensates do not co-localize with heterochromatin or nucleoli, suggesting that they are distinct and therefore silence genes through different means. Finally, making use of the corelet optogenetic system, they show that also in human cells Hes1 and TLE(Gro) can form colocalized puncta, be it upon forced nucleation through light activation.

Overall, this study makes several interesting observations concerning the potential role of liquid-liquid phase separation during transcriptional repression, which, as the authors point out, is an important yet understudied aspect of gene regulation. While droplet formation has been observed for Polycomb and HP1-associated heterochromatic regions, this study adds repressing transcription factors and the corepressors they recruit to the spectrum of repressive protein complex forming LLPS condensates. There are however several major concerns that need to be resolved prior to publication:

Major concerns

In the abstract and at the end of the manuscript the authors make several strong propositions and even generate a main figure (Fig.4) portraying how they think the repression model might work, without providing any actual data to directly support this (e.g. “We propose that repression condensates inhibit gene expression by the mechanical exclusion of transcriptional activators, coactivator complexes such as Mediator, or active chromatin (Fig. 4)”). Even though this model of activator exclusion due to repression droplet formation is attractive, the authors do not provide any data to directly support the exclusion of activators from these droplets. The authors should derive conclusions and main figures from the data presented and perhaps move such speculations and propositions to a distinct discussion section (not present in the current version of the manuscript).

Response 3.1: We have extensively rewritten the paper to provide a more modest presentation of the facts and interpretations.

Their methods section does not provide the sufficient information to assess some of experiments done in this manuscript or to potentially reproduce it. E.g what has actually been done in the Human cell system is unclear and plasmid maps do not seem to be available. Did the authors use iLID domains induce the interactions and nucleation upon light stimulus or not? If they did, what does this mean for their model? The referrals to other papers don't really help and rather make this part unnecessarily difficult to assess and vague.

Response 3.2: We have expanded the methods section to clarify the experiments performed including describing the iLID domains.

The authors state "There is only a ~3-fold increase in the levels of expression as compared with endogenous Sox 1/2/3 products (Extended Data Fig. 1)." While it is true that there is only a 3-fold increase compared to the endogenous Sox 1/2/3 products, the relevant comparison is with the endogenous levels of Hes.a, which suggests a ~20 fold increase. This needs to be explicit as it could be the source of artifacts (see below).

Response 3.3: We have covered this in our response to referee 1. The 20-fold increase is primarily due to the greater number of cells that express Hes.a with the Sox1/2/3 transgene as compared with the endogenous locus.

The authors should clarify, whether they have evidence that the diffuse green fluorescent signal derived from the mutant Hes.a_deltaWRPW in Fig. 1B, which completely fills out the whole nucleus, truly reflects a lack of puncta formation in the case of this mutant protein, or might be due to a very strong signal, making it harder to discern individual spots of aggregated protein.

Response 3.4: We consider this to be highly unlikely since we have examined nuclei using super-resolution microscopy. Additionally, we do not see a corresponding accumulation of Gro when we overexpress Hes.a deltaWRPW. Red fluorescence from Gro should not be affected by excess green signal.

In Extended Data Figure 4 the authors suggest that repression is dependent on the formation of Hes.a/Gro puncta. "These results suggest that the formation of the Hes.a puncta, binding to both DNA and Gro, are required for repression." They indeed show that for repression both Hes.a binding to DNA and Gro are required, which is known. They did however not show the presence of these puncta near repressed genes nor a causal requirement of formation of puncta for repression to occur. Hence, the relationship between the formation of repressor droplets and transcriptional repression is only correlational and fairly weak given that DNA binding and Gro recruitment are known to be necessary for repression. It should be made more obvious to the reader that such observations constitute mere coincidence rather than evidence suggestive of specific mechanisms.

Response 3.5: Yes, we agree. These studies are strictly correlative, but probably a bit better than the majority of previous studies of transcription condensates published in all of the high-profile journals. We are a bit disheartened by the claim that our correlations are weak since we believe

that we are exceeding the standards of the field. The main point is the documentation of Hes.a condensates, which we believe to represent a sufficiently significant advance in the field to justify publication in Nat Comm. As stated earlier, we have rewritten the manuscript to emphasize this point.

Fig.3 appears as if Hes.a is the only factor that binds Gro, because Gro is only present in droplets that contain Hes.a. Yet, as for example shown in Extended Data Figure 3, also Hes.b and Hes.c have WRPW domains for the interaction with Groucho and form liquid-like condensates, as should many other transcription factors. The fact that we only see Hes.a puncta in Figure 3 but no evidence of any puncta for any of the other factors that must be present suggests that the Hes.a puncta are artefacts due to the artificially high (~20 fold up) expression levels of Hes.a (see above).

Response 3.6: Hes.a, Hes.b and Hes.c are the only genes that contain the WRPW motif in the *Ciona* proteome. Other proteins may be able to bind Gro through other interaction domains, but we suspect that there may be something particular about Hes-Gro interactions that result in these large puncta. As stated earlier, the Hes genes are not being expressed at any appreciable levels when we have performed these experiments. We have included an additional supplemental figure of Gro at a later tailbud stage where puncta that resemble what we see in our Hes overexpression experiments have formed. We suspect that this is what they look like in a truly endogenous context; our transgenic assays are likely to depict a somewhat exaggerated version.

Minor remarks:

The legend of Fig. 1 states that puncta formation depends also on DNA binding of Hes.a, even though abolishing DNA binding leads to even larger puncta, as mentioned in the main text.

Response 3.7: We have rewritten the text to make our meaning clearer. Past studies of transcriptional activators have emphasized the role of the DNA template in the formation of phase separated condensates. It is possible that something similar is happening here, but our main point is that—somewhat surprisingly—the DNA template inhibits LLPS processes.

Fig. 2 should mention in the legend what blue and red colored data points refer to (We can only guess that they represent data from the two daughter cells).

Response 3.8: We have modified this figure to make this clear.

“Such fusions are readily detected for the E22V,R28C mutant (Fig. 2C, Supplementary Video 5), but not for the wild-type protein. However, there is a progressive reduction in the number of wild-type Hes.a/Gro puncta during multiple cell cycles without a corresponding diminishment in fluorescence intensity (Fig. 2A). A possible explanation for this observation is that wild-type puncta undergo fusion events as nuclei diminish in size, creating higher concentrations of compact chromatin as compared with earlier stages of development.” Their explanation that

wild-type puncta might undergo fusion events as nuclei diminish in size is rather strange, as they showed before that puncta are abolished and reformed during each cell division: “Both the wild-type and DNA binding mutant (E22V,R28C) produce puncta that are detected throughout interphase, but are abolished during mitosis before reforming in daughter nuclei (Fig. 2A, B Supplementary Videos 3, 4).”

Response 3.9: As stated above in response to referee 2 we have rewritten the manuscript to emphasize that this is purely speculative. We also provide some support for this model by documenting fusions of the wild-type Hes.a protein upon inhibition of the nuclear actin network using latrunculin A.

The inability of repressor condensates to fuse when wt Hes.a is present, can be explained by immobilization of the droplets due to binding of target DNA. In our opinion this does not weaken the argument of LLPS being involved in transcriptional repression. “We propose that interactions between proteins containing disordered domains with those containing WD40 repeats might be a key trigger for the oligomerization of biological condensates.” Please move such speculations that are not supported by experiments to a clearly separated discussion section and clearly state that these are hypotheses.

Response 3.10: Yes, we agree with these points and have revised the manuscript accordingly.

Please make sure the figure labels are clear and readable, e.g. extended data Fig.3,

Response 3.11: We have modified this figure and others where necessary to improve readability.

Extended data Fig.4: why does it seem that Hes.a::mNG signal is mainly accumulated at the cell membranes, even for the wild type Hes.a protein? Why is the membrane protein and Hes.a protein localization shown in the same color (=green)? This makes it difficult to discern where Hes.a is actually localized.

Response 3.12: We have discussed this point in response to referee 2, and have revised the figure to try make the interpretation clearer. Sorry about the confusion.

REVIEWER COMMENTS

Reviewer #1 (Remarks to the Author):

My original major criticism was that the data presented felt preliminary given the claims being made. The authors have toned down their claims and reframed the text to more accurately fit the data being presented. Overall, these edits have led to an improved manuscript. There are still a few comments that need to be addressed prior to publication.

1. The interpretations that the authors have made that DNA “inhibits” phase separation or somehow leads to a “switch in material state” are not supported by their data. What is supported is that DNA-bound condensates is less mobile and therefore less able to bounce into another condensate. The authors should remove “inhibits” and “switch in material states” as these are not supported by the data.

One of the authors has a recent preprint that highlights the dramatic role DNA-tethering can have on LLPS phenomenon of fusion and coarsening within the crowded environment of the nucleus (<https://www.biorxiv.org/content/10.1101/2020.06.03.128561v1>). It seems like concepts discussed in this preprint or a simpler restricted diffusion model are more accurate descriptor for the role of DNA. Furthermore, given overexpression of all constructs to unknown relative levels it is impossible to know whether the DNA binding domain mutant would form these condensates under physiologic expression levels and over interpretation of these results is cautioned.

2. The authors have now re-focused attention on the potential role of DNA. Given this refocus on the role of DNA, the authors should cite Shrinivas et al 2019 Molecular Cell. Shrinivas et al explores a role for DNA as a seed and scaffold to promote heterogenous co-activator/activator phase separation at specific genomic loci to prevent spontaneous and homogenous protein phase separation at random locations throughout the nucleoplasm. Granted the current manuscript focusses on co-repressor/repressor rather phase separation, the revised focus is similar to topics discussed in this previously published paper.

3. The authors should provide a rationale and expectations for latrunculin treatment. In Feric et al 2013 (cited later in the discussion), this drug is used in *Xenopus* oocytes with nuclear diameters of 450um, where nuclear F-actin is present. Is F-actin present in the smaller nucleus of *Ciona*?

4. There is no quantification or statistics of FRAP making it difficult to compare rates in the way discussed in the text.

5. Line 93. Avoid using “critical” in the colloquial sense. That word has a specific meaning for physical/chemical systems.

Reviewer #2 (Remarks to the Author):

Most of my previous comments were satisfactorily answered. But the following points are not:

Figure 2 legend: (A) Despite the authors' response, I cannot still understand what error bars mean. (1) Explain what error bars along the x-axis mean. (2) Explain what error bars along the y-axis mean.

(3) Explain what 'standard deviation +/-100sec' means. (B) Despite the authors' response, I was not able to find 'the meaning of the black, blue and red dots' on the graph.

In Response 2.9, the authors mention Hp1::mNG gives clearer signals. However, the manuscript does not contain any data or description for HP1::mNG.

Despite responses 2.6 and 2.7, there are no clear description about numbers of experiments, examined embryos/cells in the manuscript. Make every effort to avoid 'several', 'multiple', or related words other than numbers.

A related thing is line 476, 'All described phenotypes were observed in several hundred nuclei and no meaningful embryo to embryo variation was seen'. This is not clear enough, because we cannot know what the 'meaningful embryo to embryo variation' is. 'several hundred nuclei' should be replaced by actual numbers.

Other minor points:

Line 45: Because nucleoli contain Pol I, it cannot be an extreme example of 'Pol II'-mediated regulatory processes.

Line 97: References for Brachyury, Foxa.a, and Tfap2 are necessary.

Ext.Fig1: Show scale bars. The labels, 'Hes.c;mNG' and 'Hes.cDWRPW::mNG', may indicate wrong photographs.

Reviewer #3 (Remarks to the Author):

The authors have substantially revised their manuscript to clarify and adjust their claims. We agree with the authors that their findings concerning a highly conserved repressor should be of interest to all readers of Nature Communications and support publication.

Reviewer #1 (Remarks to the Author):

Overall, these edits have led to an improved manuscript. There are still a few comments that need to be addressed prior to publication.

1. The interpretations that the authors have made that DNA “inhibits” phase separation or somehow leads to a “switch in material state” are not supported by their data. What is supported is that DNA-bound condensates is less mobile and therefore less able to bounce into another condensate. The authors should remove “inhibits” and “switch in material states” as these are not supported by the data.

We have made the requested changes in the re-revised manuscript.

2. One of the authors has a recent preprint that highlights the dramatic role DNA-tethering can have on LLPS phenomenon of fusion and coarsening within the crowded environment of the nucleus (<https://www.biorxiv.org/content/10.1101/2020.06.03.128561v1>). It seems like concepts discussed in this preprint or a simpler restricted diffusion model are more accurate descriptor for the role of DNA. Furthermore, given overexpression of all constructs to unknown relative levels it is impossible to know whether the DNA binding domain mutant would form these condensates under physiologic expression levels and over interpretation of these results is cautioned.

We agree and have modified our discussion accordingly. We have also removed the claim about “switch in material state” and removed our claims about inhibition.

3. The authors have now re-focused attention on the potential role of DNA. Given this refocus on the role of DNA, the authors should cite Shrinivas et al 2019 Molecular Cell.

Done. It is now cited in the revised manuscript.

*4. The authors should provide a rationale and expectations for latrunculin treatment. In Feric et al 2013 (cited later in the discussion), this drug is used in *Xenopus* oocytes with nuclear diameters of 450um, where nuclear F-actin is present. Is F-actin present in the smaller nucleus of *Ciona*?*

The re-revised manuscript contains an expanded description of the rationale for these experiments. We’re not sure about F-actin in *Ciona* nuclei, although it is detected in mouse oocyte nuclei (Baarlink et al 2017 Nature Cell Biology). We also raise the possibility that latrunculin disrupts the perinuclear actin.

5. There is no quantification or statistics of FRAP making it difficult to compare rates in the way discussed in the text.

We have included quantifications of the FRAP results shown in figure 6. Quantifying FRAP and obtaining support for statistical comparisons is complicated by the movements of the droplets in living embryos and variability in their size. As a result, such measurements require guesswork about their changing positions in real time. We have modified our description to emphasize these limitations and instead emphasize the value of relative measurements (see below).

6. Line 93. Avoid using “critical” in the colloquial sense. That word has a specific meaning for

physical/chemical systems.

We have made the change, as requested.

Reviewer #2 (Remarks to the Author):

Most of my previous comments were satisfactorily answered. But the following points are not:

1. Figure 2 legend: (A) Despite the authors' response, I cannot still understand what error bars mean. (1) Explain what error bars along the x-axis mean. (2) Explain what error bars along the y-axis mean. (3) Explain what 'standard deviation +/-100sec' means. (B) Despite the authors' response, I was not able to find 'the meaning of the black, blue and red dots' on the graph.

We have modified the depiction of these data and removed the x-axis error bars. Full descriptions of the new presentation for these graphs are in the new figure legends and methods. The meaning of the black blue and red dots are now clearly stated on all graphs.

2. In Response 2.9, the authors mention Hp1::mNG gives clearer signals. However, the manuscript does not contain any data or description for HP1::mNG.

We had some difficulties in obtaining a Hes.a fusion protein with a stable red fluorophore.

Hp1::mNG

Hp1::mAp

Merged

The preceding image shows colocalization of Hp1::mApple (red) and HP1::mNG (green), as expected. Hp1::mApple faithfully recapitulates its green counterpart, although not as bright.

This image shows that Hp1:mNG does not colocalize with Hes.a droplets (visualized by mApple::Gro).

We therefore conclude that Hes/Gro droplets mostly do not colocalize with Hp1 constitutive heterochromatin. The purpose of these experiments was to show that we didn't just rediscover Hp1 droplets. We could include these results as supplemental data but feel that they merely confirm our previous results.

3. Despite responses 2.6 and 2.7, there are no clear description about numbers of experiments, examined embryos/cells in the manuscript. Make every effort to avoid 'several', 'multiple', or related words other than numbers. A related thing is line 476, 'All described phenotypes were observed in several hundred nuclei and no meaningful embryo to embryo variation was seen'. This is not clear enough, because we cannot know what the 'meaningful embryo to embryo variation' is. 'several hundred nuclei' should be replaced by actual numbers.

We agree and have addressed this by taking advantage of the *Ciona* electroporation system. A new supplemental figure is pasted below. It is possible to control the levels of expression by altering the amounts of plasmid DNAs in the electroporation medium. Specifically, examined a range of fusion gene concentrations, from 20 ug to 60 ug, for the two key Hes.a proteins examined in this study: Hes.a::mNG (wild-type fusion protein) and the Hes.a::mNG Δ WRPW mutant. Beyond these conditions, there is almost no detectable fluorescence at 10 ug, and intense signal continues to be observed at 80 μ g including in the cytoplasm.

The new supplemental figure that we include in the re-revised manuscript shows that Hes.a always forms droplets at varying intensities, whereas the Hes.a Δ WRPW mutant never forms droplets. The Hes.a droplets can be seen at the lowest expression levels, even though the signal is barely detectable using high resolution imaging (Zeiss Fast AiryScan). By contrast, Hes.a Δ WRPW does not form droplets even when it is overexpressed to the highest levels achievable in this system.

We hope that these experiments address any lingering concerns about the robustness, reproducibility and overall validity of our key findings.

a

b

4. Other minor points:

Line 45: Because nucleoli contain Pol I, it cannot be an extreme example of 'Pol II'-mediated regulatory processes.

Yes, we agree that Pol I and Pol II are distinct regulatory entities, but our findings are nonetheless evocative of nucleoli biogenesis.

5. Line 97: References for Brachyury, Foxa.a, and Tfp2 are necessary.

These have all been added to the revised manuscript.

6. Ext.Fig1: Show scale bars. The labels, 'Hes.c;mNG' and 'Hes.cDWRPW::mNG', may indicate wrong photographs.

We have corrected this figure and thank the referee for calling this to our attention.

Reviewer #3 (Remarks to the Author):

The authors have substantially revised their manuscript to clarify and adjust their claims. We agree with the authors that their findings concerning a highly conserved repressor should be of interest to all readers of Nature Communications and support publication.

We thank the referee for the positive assessment of our revised manuscript.